# Application of the Bromocresol Purple Index (BCPI) to Evaluate the Effectiveness of Heating Soybeans and Their Products

**Marek Szmigielski [1], Paweł Sobczak [2,\*] , Kazimierz Zawiślak [2], Dariusz Andrejko [1], Grażyna Bielecka [3], Jolanta Rubaj [3], Jacek Mazur [2] and Małgorzata Szczepanik [4]**

[1] Department of Biological Foundations of Food and Feeds Technologies, University of Life Sciences in Lublin, 20-612 Lublin, Poland; marek.szmigielski@up.lublin.pl (M.S.); dariusz.andrejko@up.lublin.pl (D.A.)

[2] Department of Food Engineering and Machines, University of Life Sciences in Lublin, 20-612 Lublin, Poland; kazimierz.zawislak@up.lublin.pl (K.Z.); jacek.mazur@up.lublin.pl (J.M.)

[3] National Laboratory for Feedingstuffs, National Research Institute of Animal Production, 20-079 Lublin, Poland; gbielecka@clpp.lublin.pl (G.B.); jrubaj@clpp.lublin.pl (J.R.)

[4] Department of Mathematics and Computer Science Applications, University of Life Sciences in Lublin, 20-612 Lublin, Poland; malgorzata.szczepanik@up.lublin.pl

\* Correspondence: pawel.sobczak@up.lublin.pl

**Abstract:** In this study, a number of the most common methods used in assessing the efficiency of soybean heat treatment were compared. All the methods proved to be useful in assessing the efficiency of heating soybean seeds and soybean products. However, considering the sensitivity, precision, time consumed, and the effectiveness of determination of the characteristics of the samples, the use of the bromocresol purple index (BCPI) appears to be justified. The BCPI method turned out to be universal, allowing distinguishing unheated ($BCPI_{BSM} < 70$ mg·g$^{-1}$), under-heated (70 mg·g$^{-1}$ < $BCPI_{BSM}$ < 130 mg·g$^{-1}$), properly heated ($BCPI_{BSM}$ = 130–140 mg·g$^{-1}$), and over-heated samples ($BCPI_{BSM} > 140$ mg·g$^{-1}$).

**Keywords:** soybean; heat treatment; trypsin inhibitor activity; urease activity; bromocresol purple index

## 1. Introduction

The high utility value of soybean results from their nutritional value, in particular the high content of well-digestible and balanced proteins that are rich in exogenous amino acids, the abundance of oil with a significant share of polyunsaturated fatty acids (PUFAs), and the vitamins and minerals they contain [1,2].

The development of optimal technology for converting soybeans into food and fodder continues to drive much of the research [3–9]. However, the use of soybean and soybean products as components of food and fodder involves various forms and methods of their heat treatment, which is applied to reduce the activity of numerous thermolabile antinutritional factors [10,11] and to develop rheological and organoleptic properties typical of soybean-based products, while maintaining the digestibility of nutrients [12]. Assessment of the effectiveness of heating soybean or other soybean products, as well as other food and fodder, can be based on any of their properties altered as the result of heating. The most common assessment methods can be methodically divided into the following five basic groups:

- Nutritional tests;
- Methods based on the determination of the content of products resulting from thermal transformations;
- assessment of non-protein properties and thermolabile quality traits;
- microbiological assays;
- Analytical methods based on alterations of the properties of product proteins [13,14].

### 1.1. Nutritional Tests

These are some of the simplest natural methods for the evaluation of raw materials, food and fodder. The impact of analysed mixtures or their components on livestock health is a factor that can be examined with such indicators as the protein efficiency ratio (PER), biological value (BV) of proteins, net protein utilisation (NPU), and true digestibility (TD) [13].

However, the results of nutritional tests may be distorted due to the high complexity of the metabolic processes experienced by experimental animals. Thus, the interpretation of such results may be difficult to perform. Therefore, agro-food processing aims to minimise the share of nutritional experiments in favour of other less time-consuming methods with a significant correlation of the results [13,14].

### 1.2. Determination of the Heating Efficiency Using Non-Protein Components

The efficiency of heating soybean seeds can be assessed by investigating the activity of thermolabile vitamins, e.g., thiamine, riboflavin, tocopherols, and tocotrienols [15] or food transformation products such as furosine [16], $\varepsilon$-pyrrolysine [17], and lysinoalanine [18].

Substantial time consumption and labour intensity, as well as the high complication of the analytical procedure, are the main disadvantages of these determinations, affecting their usefulness in the currently implemented assessment of the effectiveness of thermal treatment [13,14].

### 1.3. Microbiological Assay

Heating is a method for the reduction in microbial contamination, the level of which can serve as a measure of the length of time the product was heated at a given temperature lethal for microbes. The length of thermal treatment may result in a certain restrain on the product shelf life. A drawback of this type of determinations associated with the microbiological assays is a lengthy procedure and susceptibility to shortcomings of the analytical method [13,14].

### 1.4. Methods Based on Alterations in the Properties of Product Proteins

Three categories of methods are most frequently distinguished within this group as follows:

- Methods based on denaturing changes in the solubility or other functional properties of proteins (water absorption, emulsion stability);
- Methods based on the measurement of the activity of enzymatic proteins or some anti-nutritional factors;
- Determinations carried out with the use of indicator compounds characterised by specific reactions with certain protein groups [13].

### 1.5. Determination of Changes in the Solubility or Differences in the Functional Properties of Proteins

One of the effects of protein heating is the change in their solubility resulting in coagulation. Hence, the degree of protein heating can be determined from their solubility in water (NSI) or water solutions [19,20].

A shortcoming of these methods is the high labour intensity and significant time consumption, as well as the complexity and multistage character of the analytical procedure [13].

### 1.6. Analysis of Changes in Enzymatic Activity

The activity of enzymes contained in soybean seeds depends on the thermal denaturation of enzymatic proteins. In practice, such assays are based on urease activity (UA) or lipoxygenases activity [21–24].

Their practical application is problematic due to the time-consuming and labour-intensive character of the analyses and the risk of using varieties with genetically lowered or reduced activity of these enzymes [13].

### 1.7. Biochemical Assays

This group of methods is based on the determination of the activity of thermolabile anti-nutritional compounds present in soybean seeds, primarily the trypsin inhibitory activity (TIA) and inhibition of the activity of hemagglutinins, called lectins.

The TIA determinations are based on the reaction of trypsin with substrates (casein or the synthetic substrate BAPA N-$\alpha$-benzoyl-DL-arginine-p-nitroanilide) [25–27].

Proteins with hemagglutinating properties undergo thermal denaturation similarly to trypsin inhibitors. The hemagglutination activity is most frequently determined using the immunosorption method with the use of monoclonal [28,29] or polyclonal antibodies [30–32]. Immunosorption methods can be applied for the determination of other soy proteins ($\beta$-conglycinin) as well [29].

### 1.8. Use of Indicator Substances

This type of method is based on diagnostic reactions carried out with selected chemical reagents that specifically bind to certain groups of the heated product, facilitating assessment of the degree of heating. Such reagents as orange-G, safranin [33], coomassie blue [34–37], and cresol red (CRI) [38–40] have long been used.

The CRI value increases as the thermal treatment of soybean seeds and soybean products advances, thus facilitating the assessment of the heating intensity [13].

The index of lysine availability (LA), which is determined based on the reaction of the sample with the FDNB (1-fluoro-2,4-dinitrobenzene) reagent, in accordance with the methodology proposed by Booth [41] and Sarkar et al. [42], can be adopted as an approach for the assessment of the efficiency of heating food or fodder, including soybean seeds and their products.

The difficulty in the implementation of the FDNB-based determinations of LA and CRI is the necessity to defat samples before the determination process, which lengthens and complicates the analytical procedure [13].

Significant progress in simplification and shortening of analytical procedures in CRI- and FDNB-based studies on the efficiency of heating soybeans and soybean products was provided by the use of bromocresol purple and bromocresol green as the active compounds. This eliminated the necessity of the laborious process of sample defatting and significantly enhanced the sensitivity of the designed methods referred to as the bromocresol purple index (BCPI) and bromocresol green index (BCGI).

These methods were developed to test the efficiency of heating of selected pulses and oilseeds, such as soybean and chickpea. The studies were preceded by an initial selection of the possible indicator substances, of which acidic solutions of bromocresol purple (5′, 5″—dibromo—3′, 3″—dimethyl phenolsulfonphthalein) turned out to be the most effective ones, while in the analysis of soybean and soybean products also bromocresol green solutions (3, 3′, 5, 5′—tetrabromo-m—cresolsulfonphthalein) proved to be useful.

The sensitivity of the method ($\chi$) was the basic criterion for comparing the test variants. It was calculated as the absolute value of the direction coefficient of linear regression formula (the result of sorption of the active substance (T), calculated in accordance with formula (1), as the function of the heating time of the seeds), taking as the basis of matching the results for raw seeds samples and seeds autoclaved at the temperature of 121 °C for 120 min [13]. The sensitivity of the method increases with the higher value of the index $\chi$.

For each of the tested variants of the working solution, i.e., bromocresol purple and bromocresol green, the sorption of the active substance (T) was calculated, separately for raw and autoclaved (at 121 °C for 120 min) seeds, as the difference in the solution content before and after the contact with the ground seed sample, sieved through a 0.20 mm mesh, measured using absorbance $E_b$ and $E_o$ (formula (1)). For each of the tested variants of the

working solution, the sorption of active substance (T) was measured for raw seeds ($T_s$) as well as autoclaved seeds ($T_a$) and calculated according to formula (1). For each of the tested variants of the working solution, the sensitivity of the analytical method ($\chi$) was calculated on the basis of $T_a$ and $T_s$ values (for autoclaved and raw seeds, respectively).

$$T = (E_o - E_b) \cdot D \cdot F \cdot E_o^{-1} m^{-1} \cdot (h/100)^{-1} (z/100)^{-1} \tag{1}$$

For the variant of the working solution with the greatest sensitivity in the seed testing, the name of bromocresol purple index (BCPI) and bromocresol green index (BCGI) was used, respectively. Additionally, the lower index in $BCPI_{BSM}$ and $BCGI_{BSM}$ indicates that the calculations were made per unit weight of protein in the seeds dry weight.

Both the bromocresol purple index (BCPI) and the bromocresol green index (BCGI) were characterized by high precision ($\pi$), sensitivity ($\chi$) and high discernability ($\rho$) of the determinations [13], making it possible to accurately assess the characteristics of the samples, in the case of which the enzyme analysis (UA) and biochemical assay (TIA) were difficult to perform (Table 1).

**Table 1.** Analytical methods developed until now that use bromocresol purple and bromocresol green [13].

| Seed Species | Name of the Active Substance | Active Substance Content (mg·cm$^{-3}$) | HCl Concentration (mol·dm$^{-3}$) |
|---|---|---|---|
| soybean | bromocresol purple | 0.13 | 0.10 |
| soybean | bromocresol green | 0.25 | 0.07 |
| chickpea | bromocresol purple | 0.14 | 0.10 |

Moreover, it was also demonstrated that the results of the tests carried out using BCPI and BCGI methods were highly interdependent with the selected quality discriminants, i.e., the urease activity—UA, trypsin inhibitor activity—TIA, cresol red index—CRI, lysine absorption—LA, grass pea neurotoxin activity—BOAA, as well as acid value (AV) and peroxide value (PV) of oils and $\alpha$ tocopherol content—AE, of soybean, grass pea, bean, and rapeseed. The high and significant correlation factors enabled us to formulate regression equations correlating the soybean, grass pea, chickpea, soybean oil, and rapeseed oil quality discriminants with the values of the developed BCPI and BCGI tests as a dependent variable [13,14]. The usefulness of some of the developed regression equations was confirmed in the studies on selected characteristics of soybean meal and chickpea meal carried out on a semi-technical scale [13].

By providing the possibility to convert the results obtained, the use of these regression equations may replace some of the labour-intensive and time-consuming analytical procedures, used so far, for testing quality discriminants by fast and effective BCPI and BCGI tests.

The practical use of the developed assessment methods (BCPI and BCGI) may include the following:

- Complementing the existing methods for the evaluation of quality discriminants of seeds and products derived from them;
- Replacement of the existing evaluation methods with the possibility of converting the results using the regression equations developed;
- Using them as a replacement for the evaluation methods used so far in testing selected quality discriminants of soybean, grass pea, bean, chickpea, and rapeseed seed and their products.

So far, the most commonly applied traditional methods of analysing soybean seeds and products, despite their labour intensity and time-consuming nature, as well as complicated analytical procedures and the necessity to incorporate expensive, specialized laboratory equipment operated by highly qualified personnel, are still widely used in practice, even though thus obtained results are sometimes inaccurate and difficult to interpret. They are

based on the activity of selected enzymes (urease activity—UA) or thermolabile antinutritional factors (trypsin inhibitor activity—TIA) [43–46], often supplemented by methods based on protein solubility [18,43,45,46]. It should be noted that the UA and TIA test methods have been well verified in soybean meal studies and the obtained results of these parameters in this type of products were sufficiently correlated [47].

It appears that, under the conditions of industrial soybean processing, the bromocresol purple index (BCPI) method should meet the demands of practical implementation. This method is the answer to the approach postulating increased sensitivity of analytical methods with simultaneous shortening and simplification of the methodology [13], as it offers high sensitivity and precision results that allow for distinguishing a wide range of samples with subtle differences in properties.

The study attempts to compare the usefulness of the BCPI method with the two evaluation methods, i.e., TIA and UA, which are currently most frequently used in the assessment of the effectiveness of soybean heating.

## 2. Materials and Methods

The study used traditional varieties of soybean, i.e., not genetically modified ones, which had been purchased at the agri-food market in Elizówka near Lublin (Poland, Lublin Province). The use of traditional varieties of soybean is in line with the strategy of the European Union food processing industry as an alternative to the genetically modified soybean, the presence of which in the amount higher than 0.9% requires a declaration on the product label (food, fodder).

Fully mature soybean seeds, with chemical composition and physical characteristics typical for this species, were used in the study. The dry matter content of the seeds constituted around 93.88% of the fresh weight. In the dry matter of the seeds, there was 36.03% of total protein, 19.75% of fat, and 5.23% of ash. The dry matter studies of total protein, fat, and ash were made in accordance with the standards recommended in the EU and Poland [48–51]. Soybeans intended for the research were divided into three tranches corresponding to the methods of thermal treatment (autoclaving, microwave heating and heating, on a semi-technical scale, with an infrared radiator), implementing the intensity variants described in detail in Table 2 and Figure 1. Infrared heating consisted of three parts (humidification, specific heating and conditioning).

The heat treatment consisted of initial humidification of the samples (to the total moisture content of 15, 17, and 20%). Afterwards, the samples were evenly distributed over the area of 1000 $cm^2$ and heated for 150 or 180 s using six ceramic infrared heaters, each with the power of 400 W, located at a distance of 5 cm from the heated seeds. The testing stand is presented in Figure 1. After heating with the infrared radiators, the samples were conditioned at room temperature or in a glass dewar for the time of 5 min and then at ambient temperature (Table 2).

Each of the described heating variants (Table 2, Figure 1) was tested, per unit weight of the protein of the seeds dry matter, for urease activity (UA) according to standard [42], and for trypsin inhibitor activity (TIA) pursuant to standard [44].

For the studies using the bromocresol purple index (BCPI) method samples of soybean seeds or soybean products were ground to pass through a sieve with 0.2 mm mesh. Samples weighing precisely 0.1 g were taken from the ground soybean and transferred to a 100 $cm^3$ conical flask. Next, 50 $cm^3$ of the working solution was added and the mixture was stirred on a magnetic stirrer for 30 min. After this time, the contents of the conical flask were transferred to a centrifuge tube and centrifuged for 5 min (at 3000 rpm). After the centrifugation, 1 $cm^3$ was taken from the clear extract and transferred into a test tube containing 20 $cm^3$ of 0.02 $mol \cdot dm^{-3}$ solution of NaOH. The obtained mixture was stirred for 10 min in a test tube, and afterwards, the absorbance at 589 nm was measured using distilled water as the reference solution ($A_b$—formula (2)). Analogously, the absorbance of the working solution was measured ($A_o$—formula (2)), wherein in the place of the

centrifuged extract, 1 cm$^3$ of bromocresol purple working solution (0.13 mg·cm$^{-3}$) was added to the test tube containing 20 cm$^3$ of 0.02 mol·dm$^{-3}$ solution of NaOH.

**Table 2.** Sample preparation method.

| Section A | | | | | |
|---|---|---|---|---|---|
| **Type of thermal treatment** | **Thermal treatment parameters** | | | | |
| | | **Time of heat treatment [s]** | | | |
| Autoclaving | Temperature of heat treatment [°C] | 112 | 600 | 1200 | 1800 |
| | | 121 | 600 | 1200 | 1800 |
| | | 134 | 600 | 1200 | 1800 |
| Microwave heating | Power of Radiation [W] | 350 | 60 | 120 | 180 |
| | | 500 | 60 | 120 | 180 |
| | | 650 | 60 | 120 | 180 |

| Section B Tests Prepared on a Semi-Technical Scale/Heating with an Infrared Radiator according to Figure 1/ | | | | | |
|---|---|---|---|---|---|
| Sample | Raw soybean seeds | Moisture content (%) | Heating time (s) | Conditioning (300 s Yes/No) | |
| 0 | | - | - | N | |
| 1 | | 15 | 150 | N | |
| 2 | | 17 | 150 | N | |
| 3 | | 20 | 150 | N | |
| 4 | | 15 | 150 | Y | |
| 5 | | 17 | 150 | Y | |
| 6 | | 20 | 150 | Y | |
| 7 | | 15 | 180 | N | |
| 8 | | 17 | 180 | N | |
| 9 | | 20 | 180 | N | |
| 10 | | 15 | 180 | Y | |
| 11 | | 17 | 180 | Y | |
| 12 | | 20 | 180 | Y | |

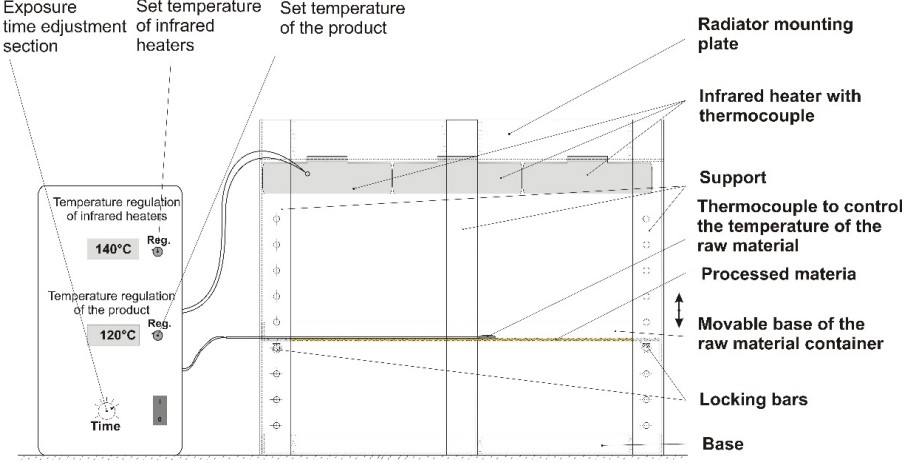

**Figure 1.** Laboratory chamber for irradiating granular materials with IR rays.

The working solution was obtained by dissolving 130 mg of bromocresol purple (5′, 5″—dibromo—3′, 3″—dimethyl phenolsulfonphthalein) in 40 cm$^3$ of NaOH solution

$(0.1 \text{ mol} \cdot \text{dm}^{-3})$ and the addition of $0.1 \text{ mol} \cdot \text{dm}^{-3}$ solution of HCl to obtain $1 \text{ dm}^3$ of the final product.

The amount of adsorbed active substance (bromocresol purple) was calculated using the formula (2) as the difference between its content in the solution before and after contact with the ground seed sample. The test result obtained according to formula (2) takes into account the conversion into the protein content in soybean dry matter, BCPI $_{BSM}$, which was possible thanks to the introduction of conversion factors.

$$\text{BCPIBSM} = \frac{(A_o - A_b) * C * V}{A_o * M * \frac{h}{100} * \frac{z}{100}} \tag{2}$$

Time consumption ($\tau$) of the analytical methods (BCPI, UA TIA) was estimated as the time required to complete one repetition of analysis per one seed sample in accordance with the established analytical procedures [13].

### 2.1. Analysis of the Raw Material Hardness

The analysis of the hardness of the raw material and soybean described in Table 2 (section B) was carried out using a TA.XT *Plus* texture analyser made by Stable Micro Systems. The measurements were performed using a compression test on a TA.XT Plus device with a 500 N head. The whole seeds, laid horizontally in a position perpendicular to the base of cotyledon split surface, were compressed to 50% of their original height with the head travel speed of 0.83 mm/s. The hardness was defined as the maximum value of the force recorded during the test and it was read from the force-displacement graph. The measurements were made at the ambient temperature of $20 \pm 1$ °C. The tests were performed in 5 repetitions.

### 2.2. Statistical Analysis of the Research Results

The test results for soybean samples, prepared according to Table 2, obtained using the BCPI$_{BSM}$, UA, and TIA methods were statistically analysed by calculating the average values (S) and standard deviation (SD) of the measurements as well as the accuracy of determination ($\pi$—this value is numerically equal to the coefficient of variation (CV) of these determinations) [52]. The precision of the determinations increases with the decreasing value of the coefficient of variation of the measurements carried out.

The experimental data (UA, BCPI$_{BSM}$ and TIA) concerning all the microwave heating and autoclaving variants (Table 2, section A) were compiled as points representing (BCPI$_{BSM}$, TIA% act) and (BCPI$_{BSM}$, UA% act). The independent variable for each of these points is the mean value of BCPI$_{BSM}$, and the dependent variable is the mean value of TIA% act or UA% act, calculated for each of the heating variants (autoclaving, microwave heating) as the ratio of the mean value of UA or TIA to raw soybeans activity. The experimental data developed in this way was approximated (using the least-squares method) by the curves whose course was characterized by the best possible matching ($R^2$) to the experimental points [52].

Taking into account the average values (S) and standard deviations (SD) the significance of differences between average values of the measurements for each of the methods (BCPI, UA TIA) were assessed (for a 5% significance level). The significance of differences between average values obtained using each of the methods, i.e., BCPI$_{BSM}$, UA, and TIA, for the samples prepared in accordance with Table 2 section B, formed the basis for distinguishing the results of measurements ($\rho$), expressed as the percentage of relevant relations in respect to the total verified relations [52]. The higher the value of $\rho$ (maximally 100%), the greater the distinguishability of the research results.

The experimental data (BCPI, TIA, UA) for autoclaved and microwave heated soybeans statistically compiled by performing a one-way analysis of variance test at a 5% significance level [13,52]. Detailed mean differences were examined with the Tukey test.

The significance of the differences between the means of each of the features within each of the methods and variants of heating became the basis for the distinguishability of

measurement results ($\rho$), expressed as a percentage of the significant relationships with respect to the total number of verified relations [13].

## 3. Results and Discussion

Trypsin inhibitor activity (TIA) of the soybean seeds reference sample, i.e., not heat-treated, was at the level typical for this kind of material (Tables 3 and 4), which was also in line with the data from literature of the subject [2,53–59].

**Table 3.** The results obtained by implementing the bromocresol purple index (BCPI), urease activity (UA) and trypsin inhibitor activity (TIA) methods in the assessment of soybean seeds samples.

| | BCPI$_{BSM}$ [mg·g$^{-1}$$_{BSM}$] | | | UA [mg$_N$ g$^{-1}$·min$^{-1}$] (30 °C) | | | TIA [mg·g$^{-1}$] | |
|---|---|---|---|---|---|---|---|---|
| Sample Acc. to Table 2 Section B | Value | $\pi$ (%) | Sample Acc. to Table 2 section B | Value | $\pi$ (%) | Sample Acc. to Table 2 section B | Value | $\pi$ (%) |
| 0 | a)65.07 ± 1.00 | 1.54 | 12 | a,b,c,d,e,f,g,h,i,j)0.05 ± 0.02 | 40.00 | 12 | a,b,c,d,e,f,g,h,i,j,k)3.20 ± 0.20 | 6.25 |
| 1 | b)120.22 ± 1.32 | 1.10 | 11 | b,c,d,e,f,g,h,i,j)0.05 ± 0.02 | 40.00 | 11 | b,c,d,e,f,g,h,i,j,k)3.20 ± 0.20 | 6.25 |
| 4 | c)125.65 ± 1.20 | 0.95 | 10 | c,d,e,f,g,h,i,j)0.05 ± 0.02 | 40.00 | 10 | c,d,e,f,g,h,i,j,k)3.20 ± 0.20 | 6.25 |
| 2 | d)130.48 ± 1.34 | 1.03 | 9 | d,e,f,g,h,i,j)0.05 ± 0.02 | 40.00 | 9 | d,e,f,g,h,i,j,k)3.20 ± 0.20 | 6.25 |
| 3 | e)135.50 ± 1.54 | 1.14 | 8 | e,f,g,h,i,j)0.05 ± 0.02 | 40.00 | 8 | e,f,g,h,i,j,k)3.20 ± 0.20 | 6.25 |
| 7 | f)139.26 ± 1.27 | 0.91 | 7 | f,g,h,i,j)0.05 ± 0.02 | 40.00 | 7 | f,g,h,i,j,k)3.20 ± 0.20 | 6.25 |
| 8 | g)143.11 ± 1.22 | 0.85 | 6 | g,h,i,j)0.05 ± 0.02 | 40.00 | 6 | g,h,i,j,k)3.20 ± 0.20 | 6.25 |
| 9 | g,h)145.67 ± 1.54 | 1.06 | 5 | h,i,j)0.05 ± 0.02 | 40.00 | 5 | h,i,j,k)3.20 ± 0.20 | 6.25 |
| 5 | g,h,i)146.22 ± 1.66 | 1.14 | 4 | i,j)0.05 ± 0.02 | 40.00 | 4 | i,j,k)3.20 ± 0.20 | 6.25 |
| 6 | j)151.33 ± 1.44 | 0.95 | 3 | j)0.05 ± 0.02 | 40.00 | 3 | j,k)3.40 ± 0.20 | 5.88 |
| 10 | k)156.69$^j$ ± 1.72 | 1.10 | 2 | k)0.14 ± 0.03 | 21.43 | 2 | k)3.60 ± 0.30 | 8.33 |
| 11 | k,l)158.19$^j$ ± 2.20 | 1.40 | 1 | l)1.70 ± 0.17 | 10.00 | 1 | l)6.40 ± 0.50 | 7.81 |
| 12 | m)162.22$^k$ ± 1.49 | 0.92 | 0 | m)7.52 ± 0.21 | 2.79 | 0 | m)23.10 ± 0.21 | 3.29 |
| | $\pi_{min}$= 1.54 | | | $\pi_{min}$= 40.00 | | | $\pi_{min}$= 8.33 | |
| | $\tau$ [h] = 1.5 | | | $\tau$ [h] = 3.0 | | | $\tau$ [h] = 4.5 | |
| | $\rho$ [%] = 94.87 | | | $\rho$ [%] = 53.85 | | | $\rho$ [%] = 53.85 | |

Values marked in the columns with the same letter are not significantly different ($p < 0.05$).

**Table 4.** Urease activity (UA) [mg$_N$·g$^{-1}$·min$^{-1}$] (30 °C) for soybean seeds heated by the means of electromagnetic radiation of 2450 MHz frequency.

| Urease Activity UA [mg$_N$·g$^{-1}$·min$^{-1}$] (30 °C) | | | |
|---|---|---|---|
| Without thermal treatment | 7.52 ± 0.21 (2.79) /100.00/ | | |
| Type of thermal treatment | Thermal treatment parameters | | |
| | Power of Radiation [W] | Time of thermal treatment [s] | |
| | | 60 | 120 | 180 |
| Microwave heating | 350 | 4.24 ± 0.25 (5.90)/56.38/ | 2.54 ± 0.30 (11.81)/33.78/ | 0.27 ± 0.09 (33.33)/3.59/ |
| | 500 | 2.21 ± 0.22 (9.95)/29.39/ | 0.21 ± 0.09 (42.86)/2.79/ | 0.12 ± 0.05 (41.67)/1.60/ |
| | 650 | 0.23 ± 0.19 (82.61)/3.06/ | 0.13 ± 0.06 (46.15)/1.73/ | 0.09 ± 0.04 (44.44)/1.20/ |

In ( ) brackets—variability coefficients for the obtained results—as a %, in // brackets—percentage of the trait value of UA in reference to the level recorded for the raw seeds. LSD = 0.36; $\rho$ = 66%

Heating the seeds in accordance with the methodology described in Table 2 resulted in a significant reduction in TIA in the samples. Except for the reference sample and sample 1,

the remaining ones were characterized by a trace level of TIA, viz. the properties indicating their fitness for consumption.

It is assumed that the acceptable level of TIA in food and fodder (expressed as mg of trypsin inhibitor per gram of protein) is typically estimated at the level of the tenth part of the protein percentage of the product [1]. Generally, TIA of heat-treated soybean, determined according to the accepted methodology, is never below a certain trace level, which is most often explained as the result of a varied thermal resistance of trypsin inhibitors present in soybean seeds (Kunitz-type Inhibitor—KTI, and Bowman–Birck Inhibitor—BBI) [10]. Most of the soybean samples were characterised by a similar level of TIA. This resulted in low distinguishability ($\rho$ = 53.85%—Table 3), despite the differences in the method of sample preparation, i.e., varied length and temperature of the treatment. A characteristic feature of soybean subject to heat treatment is that sometimes the precision of assessment carried out using the TIA method is lowered. This is particularly common in the case of samples qualified as suitable for consumption, for which sometimes the measured value is comparable with the standard deviation of the measurement (Table 3). Decreased distinguishability and reduced precision of the TIA determinations can make it difficult to qualify soybean products correctly, particularly those the quality of which might be sometimes reduced due to excessive heat treatment [12]. Analogous results were obtained in the course of studies on soybean seeds and soybean meal treated with different heating methods [13], and in the works on microwave-heated soybean seeds [47]. The results of studies carried out on soybean flour samples obtained from Nigerian soybean varieties [57] may also be interpreted similarly.

Raw soybean seeds, constituting the reference sample, were characterized by levels of urease activity (UA—Table 3) typical for this material, which is analogous to the data from the literature of the subject [13]. Subjecting soybean seeds to heat treatment in accordance with the method described in detail in Table 2 led to a change in the characteristics of these samples, including a significant decrease in the urease enzyme activity. Most samples of tested soybean seeds were characterised by zero or trace amounts of UA, thus achieving consumption usefulness according to the guidelines in the literature of the subject [1]. However, the reduction in urease activity to zero or trace levels was accompanied by a decreased precision of the determinations as well as by similarity of physicochemical properties, observed as a similar or identical value of urease activity (UA), which resulted in a limited distinguishability of the samples ($\rho$ = 53.85%—Table 3). The results obtained indicate the limited usefulness of the UA method for evaluating soybean seeds or soybean products characterised by a significant reduction in the urease activity, which may lead to an incorrect classification of over-heated material with reduced digestibility of proteins [12]. Analogous conclusions were formulated based on the previous studies by Szmigielski [13] or can be drawn by analysing the results of studies conducted by Anozie et al., Caprita et al., Varga-Visi et al. and Purushotam et al. [53–59].

The value of bromocresol purple index (BCPI$_{BSM}$) for raw, unheated soybean seeds was at a typical level, similar to the literature data, viz. it did not exceed 70 mg·g$^{-1}$ [13].

The measured values for the treated seed samples varied significantly, indicating a large range of BCPI$_{BSM}$ variations. The values were often over two times greater than the values recorded for the raw seeds. Moreover, the results obtained are characterised by high and consistent accuracy ($\pi$ factor did not exceed 3% of the measured value and remained at a similar level regardless of the thermal treatment of samples—Table 3).

The results of the study by Szmigielski [13], obtained for soybean seeds, soybean meal, and soybean exudates, were also characterised by a similar, or even greater, range of change of BCPI$_{BSM}$, as well as high and consistent accuracy to within a few percent. The broad range and high precision of the BCPI$_{BSM}$ measurements implied high distinguishability of the samples ($\rho$ = 94.87), which indicates that the sensitivity of the method applied is sufficient to accentuate the subtle differences in physicochemical properties of most of the samples prepared (Table 3). A similar, or even higher value of distinguishability ($\rho$) was obtained in the aforementioned work by Szmigielski [13] as well. For most of the

heat-treated soybean seeds, the result obtained exceeded 130 mg·g$^{-1}$, which, based on the comparative studies by Szmigielski [13], was accepted as the BCPI$_{BSM}$ level corresponding to consumer suitability of soybean seeds. The results obtained so far (Table 3) indicate that the range of changes in the BCPI$_{BSM}$ may also be an argument for the application of this method for evaluating over-heated samples of soybean and soybean products, the usefulness of which is reduced by the limited digestibility of nutrients [12].

The test results presented in Table 4 were obtained by determining the urease activity (UA) for microwave-heated soybeans. The reference sample, containing raw soybean, showed the highest measurement value (7.52 mg$_N$·g$^{-1}$ for 30 °C), while the smallest (0.09 mg$_N$·g$^{-1}$ for 30 °C), representing 1.2% of the UA level of raw soybean, was measured in the case of the test sample heated for 3 min with 650W radiation. Similar residual values of UA, with low precision of these determinations, were also measured in the case of seeds that had been microwave-heated with radiation of 500W for two and three min and radiation of 650W for one, two and three min. These data may also be the outcome of low discrimination of these trials.

Similar conclusions were presented in the earlier work [13].

Table 5 shows the experimental data obtained after testing the trypsin inhibitor activity (TIA) of microwave-heated soybean. Raw soybean had the highest measurement value (22.36 mg·g$^{-1}$), and the lowest (2.78 mg·g$^{-1}$), representing 12.43% of the TIA level of raw seeds, was observed in test samples heated by microwave radiation for 3 min with 650 W power. The reduction in the TIA measurement value is accompanied by a reduction in the precision of determinations, which may also result in reduced distinguishability of the samples.

Similar conclusions were presented in the earlier work [13].

The experimental data obtained by the bromocresol purple index (BCPI) method for microwave-heated soybean are shown in Table 6. The lowest BCPI value (65.07 mg·g$_{BSM}$$^{-1}$) was measured for raw soybean, while the highest (130.11 mg·g$_{BSM}$$^{-1}$), constituting 199.95% of the BCPI level of raw seeds, was observed in seeds heated by microwave radiation with a power of 650W for three min. The obtained BCPI results are characterized by high, uniform precision of the determinations, which was independent of the heating intensity the samples were subjected to and may indicate good distinguishability of samples.

**Table 5.** Trypsin inhibitor activity (TIA) [mg·g$^{-1}$] for soybean seeds heated by the means of electromagnetic radiation of 2450 MHz frequency.

| Trypsin Inhibitor Activity (TIA) [mg·g$^{-1}$] | | | |
|---|---|---|---|
| Without thermal treatment | 22.36 ± 0.53 (2.37)/100.00/ | | |
| Type of thermal treatment | Power of Radiation [W] | Time of thermal treatment [s] | | |
| | | 60 | 120 | 180 |
| Microwave heating | 350 | 19.02 ± 0.68 (3.58)/85.06/ | 12.64 ± 0.55 (4.35)/56.53/ | 4.22 ± 0.72 (17.06)/18.87/ |
| | 500 | 16.26 ± 0.80 (4.92)/72.72/ | 8.06 ± 0.47 (5.83)/36.05/ | 3.41 ± 0.28 (8.21)/15.25/ |
| | 650 | 9.13 ± 0.60 (6.57)/40.83/ | 4.33 ± 0.45 (10.39)/19.36/ | 2.78 ± 0.33 (11.87)/12.43/ |

In ( ) brackets—variability coefficients for the obtained results—as a %, in // brackets—percentage of the trait value of TIA in reference to the level recorded for the raw seeds. LSD = 1.20; ρ = 87%

**Table 6.** Bromocresol purple index j [mg·$g_{BSM}^{-1}$] for soybean seeds heated by the means of electromagnetic radiation of 2450 MHz frequency.

| Bromocresol Purple Index (BCPI) [mg·$g_{BSM}^{-1}$] | | | | |
|---|---|---|---|---|
| Without thermal treatment | | 65.07 ± 1.00 (1.54)/100.00/ | | |
| Type of thermal treatment | Power of Radiation [W] | Time of thermal treatment [s] | | |
| | | 60 | 120 | 180 |
| Microwave heating | 350 | 73.92 ± 1.87 (2.53)/113.60/ | 86.66 ± 2.14 (2.47)/133.18/ | 94.18 ± 1.76 (1.87)/144.74/ |
| | 500 | 89.12 ± 2.04 (2.29)/136.96/ | 103.53 ± 1.97 (1.90)/159.11/ | 122.52 ± 1.60 (1.31)/188.29/ |
| | 650 | 102.62 ± 2.22 (2.16)/157.71/ | 117.56 ± 2.26 (1.92)/180.67/ | 130.11 ± 1.84 (1.41)/199.95/ |

In ( ) brackets—variability coefficients for the obtained results—as a %, in // brackets—percentage of the trait value of BCPI in reference to the level recorded for the raw seeds. LSD = 4.04; ρ = 95%

Similar conclusions were presented in the earlier work [13].

The experimental data in Table 7 shows the results of urease activity (UA) determinations for autoclaved soybean. The raw seeds, which are the reference sample, were characterized by the highest urease activity (7.52 $mg_N·g^{-1}$ for 30 °C). Autoclaving at 134 °C for 10, 20 and 30 min, as well as for 30 min at 121 °C, resulted in the reduction in the UA value to a residual level, which was accompanied by decreased precision of these determinations, which may also imply a reduced distinguishability between these test samples (Table 7). A similar conclusion was presented in the earlier study [13].

**Table 7.** Urease activity (UA [$mg_N·g^{-1}·min^{-1}$] (30 °C) for autoclaved soybean seeds.

| Urease Activity UA [$mg_N·g^{-1}·min^{-1}$] (30 °C) | | | | |
|---|---|---|---|---|
| Without thermal treatment | | 7.52 ± 0.21 (2.79)/100.00/ | | |
| Type of thermal treatment | Thermal treatment parameters | | | |
| | Temperature of thermal treatment [°C] | Time of thermal treatment [min] | | |
| | | 10 | 20 | 30 |
| Autoclaving | 112 | 5.76 ± 0.15 (2.60)/76.60/ | 4.72 ± 0.13 (2.75)/62.77/ | 2.24 ± 0.09 (4.02)/29.79/ |
| | 121 | 4.42 ± 0.17 (3.85)/58.78/ | 2.57 ± 0.10 (3.89)/34.18/ | 0.15 ± 0.07 (46.67)/1.99/ |
| | 134 | 0.27 ± 0.11 (40.74)/3.59/ | 0.19 ± 0.08 (42.11)/2.53/ | 0.09 ± 0.05 (55.55)/1.20/ |

In ( ) brackets—variability coefficients for the obtained results—as a %, in // brackets—percentage of the trait value of UA in reference to the level recorded for the raw seeds. LSD = 0.26; ρ = 86%

Table 8 presents the experimental data obtained for the measurements of trypsin inhibitor activity (TIA) of autoclaved soybean. The highest TIA value (22.36 mg·$g^{-1}$) was recorded for raw soybean, which constitutes the reference sample. The lowest value of trypsin inhibitor activity (2.85 mg·$g^{-1}$) was observed in the case of seeds autoclaved for 30 min at 134 °C. These seeds were also characterized by reduced precision of determinations (Table 8 ). At least two samples autoclaved for 30 min at 121oC and for 20 min at

134 °C (Table 8) also provided reduced precision of determinations and a relatively low measurement value of trypsin inhibitor activity (TIA). The reduction in the measurement value happening simultaneously with the reduction in precision of the determinations could make interpreting the test results difficult and reduce the differentiation of samples. Similar conclusions were presented in the earlier study [13].

**Table 8.** Trypsin inhibitor activity (TIA) [mg·g$^{-1}$] for autoclaved soybean seeds.

| Trypsin Inhibitor Activity (TIA) [mg·g$^{-1}$] | | | | |
|---|---|---|---|---|
| Without heat treatment | | 22.36 ± 0.53 (2.37)/100.00/ | | |
| Type of thermal treatment | Temperature of thermal treatment [°C] | Time of thermal treatment [min] | | |
| | | 10 | 20 | 30 |
| Autoclaving | 112 | 18.85 ± 0.60 (3.18)/84.30/ | 17.45 ± 0.38 (2.18)/78.04/ | 3.36 ± 0.40 (11.90)/15.02/ |
| | 121 | 10.09 ± 0.45 (4.46)/45.13/ | 6.18 ± 0.43 (6.96)/27.64/ | 3.05 ± 0.39 (12.79)/13.64/ |
| | 134 | 3.26 ± 0.32 (9.82)/14.58/ | 3.09 ± 0.34 (11.00)/13.82/ | 2.85 ± 0.36 (12.63)/12.75/ |

In ( ) brackets—variability coefficients for the obtained results—as a %, in // brackets—percentage of the trait value of TIA in reference to the level recorded for the raw seeds. LSD = 0.90; ρ = 78%

Table 9 summarizes the experimental data obtained by the bromocresol purple index (BCPI) method for autoclaved soybeans. Raw soybean of the reference sample had the lowest BCPI value (65.07 mg· g$_{BSM}$$^{-1}$), while the highest values were recorded for those autoclaved for 30 min at 134 °C (126.77 mg · g$_{BSM}$$^{-1}$). The test results were characterized by high, uniform precision, which was independent of the intensity of autoclaving and may indicate a good method of differentiation of the samples.

Similar conclusions were formulated in the earlier study on similar topics [13].

Approximation of the experimental points (BCPI, UA%) and (BCPI, TIA%) for autoclaved and microwave-heated seeds (Tables 4–9) provided curves that can be used to convert these data and replace complex analytical methods (UA, TIA) with a faster and easier to perform BCPI method. The credibility of fitting these curves in the description of the experimental data is confirmed by the high coefficients of determination R$^2$ (Table 10). Similar results were obtained in the earlier studies by Szmigielski [13].

**Table 9.** Bromocresol purple index [mg·g$_{BSM}$$^{-1}$] for autoclaved soybean seeds.

| Bromocresol Purple Index (BCPI) [mg·g$_{BSM}$$^{-1}$] | | | | |
|---|---|---|---|---|
| Without heat treatment | | 65.07 ± 1.00 (1.54)/100.00/ | | |
| Type of thermal treatment | Temperature of thermal treatment [°C] | Time of thermal treatment [min] | | |
| | | 10 | 20 | 30 |
| Autoclaving | 112 | 68.12 ± 2.02 (2.97)/104.69/ | 70.86 ± 1.63 (2.30)/108.90/ | 83.73 ± 1.58 (1.89)/128.68/ |
| | 121 | 75.22 ± 1.54 (2.05)/115.60/ | 80.04 ± 2.32 (2.90)/123.01/ | 91.12 ± 1.34 (1.47)/140.03/ |
| | 134 | 93.03 ± 1.72 (1.53)/142.97/ | 102.62 ± 1.92 (1.87)/157.71/ | 126.77 ± 1.44 (1.14)/194.82/ |

In ( ) brackets—variability coefficients for the obtained results—as a %, in // brackets—percentage of the trait value of BCPI in reference to the level recorded for the raw seeds LSD = 3.58; ρ = 93%

**Table 10.** Urease activity (UA) and trypsin inhibitor activity (TIA) as a function of the bromocresol purple index (BCPI) for autoclaved soybean seeds and samples heated by means of electromagnetic radiation of 2450 MHz frequency.

| Independent Variable | Dependent Variable | Type of Equation | Equation | Determination Coefficient |
|---|---|---|---|---|
| | | Autoclaved soybean seeds | | |
| $BCPI_{BSM}$ | UA% | polynomial °II | $UA = 0.0005(BCPI)^2 - 0.102(BCPI) + 5.5936$ | $R^2 = 0.9694$ |
| | UA% | exponential | $UA = 148.61 \times e^{-0.081(BCPI)}$ | $R^2 = 0.7713$ |
| | TIA% | polynomial °II | $TIA = 0.0004(BCPI)^2 - 0.0915(BCPI) + 5.0041$ | $R^2 = 0.7481$ |
| | TIA% | exponential | $TIA = 4.1582 \times e^{-0.032(BCPI)}$ | $R^2 = 0.5744$ |
| | | Soybean seeds heated by means of electromagnetic radiation of 2450 MHz frequency | | |
| $BCPI_{BSM}$ | UA% | polynomial °II | $UA = 0.0003(BCPI)^2 - 0.0767(BCPI) + 4.4582$ | $R^2 = 0.9278$ |
| | UA% | exponential | $UA = 94.542 \times e^{-0.072(BCPI)}$ | $R^2 = 0.8239$ |
| | UA% | logarithmic | $UA = -0.949\ln(BCPI) + 4.5274$ | $R^2 = 0.7453$ |
| | TIA% | polynomial °II | $TIA = 0.0002(BCPI)^2 - 0.0607(BCPI) + 4.0925$ | $R^2 = 0.9213$ |
| | TIA% | exponential | $TIA = 12.405 \times e^{-0.036(BCPI)}$ | $R^2 = 0.8857$ |
| | TIA% | logarithmic | $TIA = -1.319\ln(BCPI) + 6.4694$ | $R^2 = 0.8808$ |

The similarity of the regression equations of a specific type TIA% = f (BCPI) and UA% = f (BCPI), including the experimental data for autoclaved and microwave-heated soybean, indicates the universality of this type of curves and their applicability to the quality control of soybean products obtained on an industrial scale, e.g., granules and soybean extrudates. Similar conclusions were formulated in the earlier study on similar topics [13].

Moreover, considering the time needed for the analysis, the implementation of the BCPI method is also the most favourable option in assessing the effectiveness of heat treatment of soybean seeds and products (Table 3).

The measurement method using the BCPI index can be applied to various agricultural materials. In the food industry, soybean is processed in various forms.

The conditioning process, i.e., retaining soybean at a higher temperature, resulted in decreased seed hardness as compared to the control test and non-conditioned seed (Figures 2 and 3). This is due to prolonged exposure to temperature without removing the water from the product at the same time. This process also positively affected the reduction in antinutritional compounds in soybean. In the research by Lara et al. [60], soybean seeds were subjected to an IR treatment in order to perform the scalding process that resulted in the reduction in weight and hardness. After 100 s of heating, the lowest hardness was obtained at the level of the scalding process. Heating for less than 100 s did not produce the expected softening results. Andrejko et al. [61] conducted studies on the effects of microwave processing on soybean. These studies determined the stress values inside the seed during humidification in the case of seeds micronized at 180 °C for 120 s and those without heat treatment. The results show a significant impact of micronization on the seed structure, as the stress values were ten times lower than those of the control samples. Infrared radiation treatment was also used in the case of African legumes [62]. As reported by the study, water is absorbed faster, which resulted in better viscosity of the pastes obtained. The resulting structural changes caused by infrared heating

of various grain legumes lead to the interactions of the seeds biomolecules that depend on the moisture content.

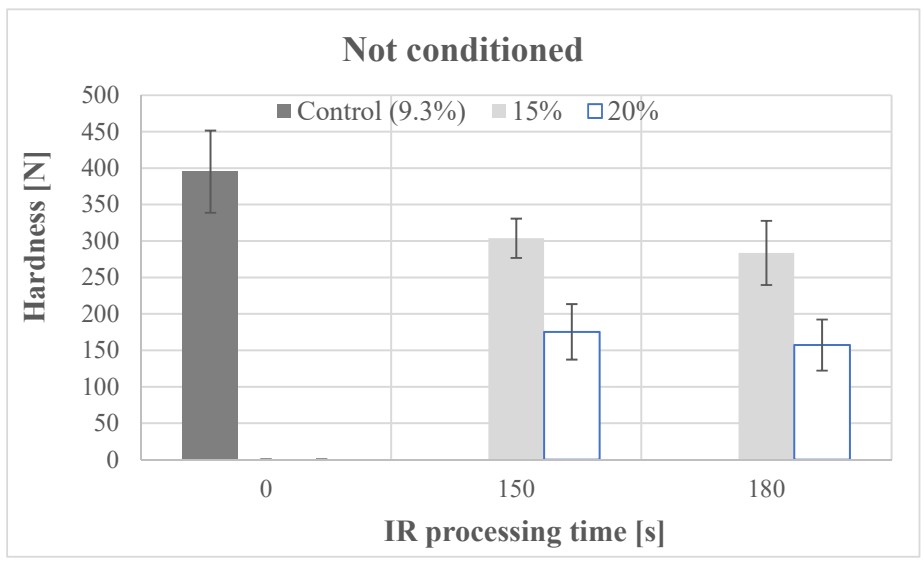

**Figure 2.** Results of the soybean hardness measurements for not preconditioned seeds after microwave treatment.

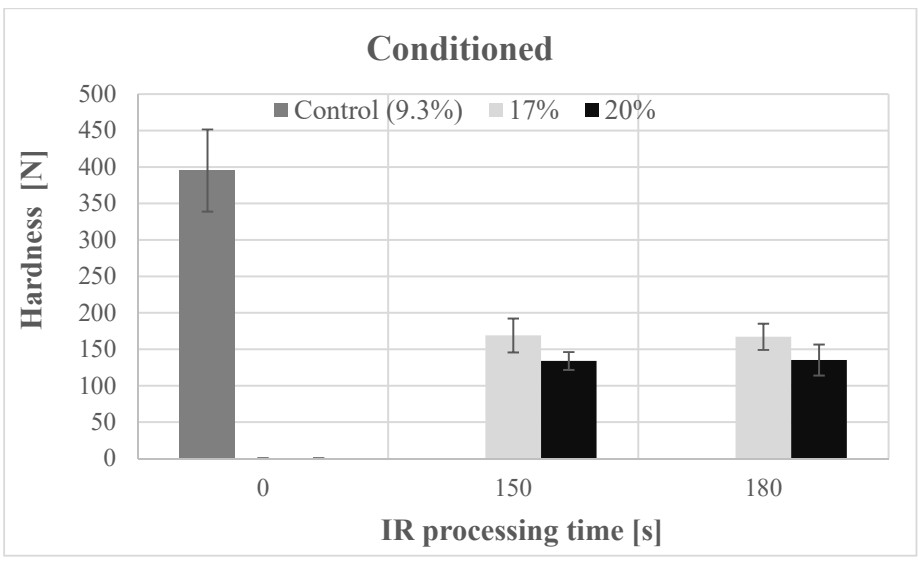

**Figure 3.** Results of the soybean hardness measurements for preconditioned seeds.

## 4. Conclusions

All the methods used, i.e., BCPI, UA, and TIA, proved useful in assessing the efficiency of soybean seeds heat treatment. However, considering the sensitivity, precision, the time necessary for performing the analysis, and the efficiency in distinguishing the characteristics of the samples, the use of the bromocresol purple index (BCPI) method seems to be the most rational.

Implementation of the bromocresol purple index (BCPI) method in soybean processing technologies may be carried out as follows:

- Complementation to the previously used methods of assessment, viz. UA, TIA, and PDI;
- Replacement of the previous methods (UA, TIA PDI), with the possibility of recalculating the results obtained in accordance with developed algorithms.

The BCPI method proved to be universal, making it possible to distinguish between the following: unheated samples (BCPI < 70 mg·g$^{-1}$), under-heated samples (70 mg·g$^{-1}$ < BCPI < 130 mg·g$^{-1}$), properly heated ones (BCPI = 130–140 mg·g$^{-1}$), and over-heated ones (BCPI > 140 mg·g$^{-1}$).

**Author Contributions:** Conceptualization, data curation, investigation, methodology, M.S. (Marek Szmigielski); Writing—original draft, investigation, P.S.; Visualization, supervision, conceptualization, K.Z.; Visualization, supervision, D.A.; Investigation, G.B.; Investigation, J.R.; Data curation, formal analysis, J.M.; Statistical processing of experimental data, M.S. (Małgorzata Szczepanik). All authors have read and agreed to the published version of the manuscript.

**Funding:** This research received no external funding.

**Institutional Review Board Statement:** Not applicable.

**Informed Consent Statement:** Not applicable.

**Data Availability Statement:** Not applicable.

**Conflicts of Interest:** The authors declare no conflict of interest.

## Nomenclature

T—sorption of the active substance on the seed surface (per unit weight of protein in the seeds dry weight (mg·gBSM$^{-1}$),

$E_o$—absorbance of the solution after the addition of 1cm$^3$ of working solution to 20 cm$^3$ of 0.02 mol·dm$^{-3}$ NaOH solution,

$E_b$—absorbance of the solution after the addition of 1 cm$^3$ of the extract (after centrifugation) to 20 cm$^3$ of 0.02 mol·dm$^{-3}$ NaOH solution,

D—concentration of the working solution (mg·cm$^3$),

F—the volume of the working solution added to the seed sample (cm$^3$),

M—the weight of seeds sample (g),

h—seeds dry matter content (%),

z—protein content in the seeds dry matter (%),

UA—urease activity,

TIA—trypsin inhibitor activity,

CRI—cresol red index,

LA—lysine availability,

BOAA—the activity of grass pea neurotoxin,

AV—acid value,

PV—peroxide value of oils and α tocopherol content,

BCGI—bromocresol green index,

π—precision,

χ—sensitivity,

ρ—discernability,

$BCPI_{BSM}$—bromocresol purple index (per unit weight of protein in the seeds dry weight—mg·g$_{BSM}$ $^{-1}$),

$A_o$—absorbance of the solution obtained after adding 1 cm$^3$ of working solution (0.13 mg·cm$^{-3}$) to 20 cm$^3$ of 0.02 mol·dm$^{-3}$ solution of NaOH,

$A_b$—absorbance of the solution obtained after adding 1 cm$^3$ of the extract (after centrifugation) to 20 cm$^3$ of 0.02 mol·dm$^{-3}$ solution of NaOH,

C—concentration of the working solution (0.13 mg·cm$^{-3}$),

V—the volume of the working solution (50 cm$^3$),

M—the weight of the seeds test sample,

h—seeds dry matter content (%),

z—protein content in the seeds dry matter (%),

(h/100)·(z/100)/h—% of seeds dry matter content.

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
