# Peer review of "Application of the Bromocresol Purple Index (BCPI) to Evaluate the Effectiveness of Heating Soybeans and Their Products"

_sustainability, doi:10.3390/su14031872_

Round 1

Reviewer 1 Report

The subject of article entitled “Application of the bromocresole purple index (BCPI) to evaluate the effectiveness of heating soybeans and their products” is interesting, and the article is well presented. However, there are some major issues to be addressed before accepted for publications. My comments are presented below:

Line 334. Authors wrote that in table 2 was described hardness the raw material and soybean. In table 2 Authors did not included this. Authors should correct that. Meybe Authors things about figure 2?

22 references are more than 22 years old. authors should provide more recent literature.

Author Response

Reply to the reviewers                               

We would like to thank the Reviewers for his interest in our work and for helpful comments that will greatly improve the manuscript and we have tried to do our best to respond to the points raised. All comments of reviewers were taken into account and appropriate corrections were introduced in the text.              

  Reviwer 1:

  1.      “Line 334 (…)”Corrected in the text by correcting the error.
  2.       “22 references are more than 22years old.(…)”                                 
    We are grateful to the journal "Sustainability" (Editor, Reviewers) for the opportunity to present a broad context of research on the issues of assessing the heating efficiency of soybeans and other food and feed ingredients. The study abandoned a lapidary, abbreviated introduction in favor of a broad introduction that covered the entirety of this type of research, including the development of these analytical techniques, often going back to the past and based on the original works of precursors. The adopted formula for this part of the work, as a consequence, resulted in reaching for original, historical publications that influenced the compilation of the literature.                 In the experimental part of the project of our publication, in accordance with the canons, mainly current literature was used.

English was moderated.

Reviewer 2 Report

Interesting topics covered in this paper “Application of the bromocresole purple index (BCPI) to evaluate the effectiveness of heating soybeans and their products”. How to quickly and effectively evaluate whether soybeans are overheated is useful, this paper is suitable for MDPI Sustainability. Authors should clarify some points:
1. Is there any practical experience in evaluating the effectiveness of heating soybeans by BCPI?
2. Line 90-Line 272, The content of Introduction is a bit long, can it be simplified?
3. Table 4, Table 5, Table 6, The content of these tables is substantial, but confusing, it is best to redesign.
4. The data comparison of BCPI, UA and TIA involved in this paper should have significant analysis.

Author Response

We would like to thank the Reviewers for his interest in our work and for helpful comments that will greatly improve the manuscript and we have tried to do our best to respond to the points raised. All comments of reviewers were taken into account and appropriate corrections were introduced in the text.

Reviwer 2: 

  1.      Is there any practical experience in evaluating the effectiveness of heating soybeans by BCPI?

Yes, the analytical technique called the bromocresol purple index (BCPI) has been successfully applied to soybean extrudates on a pilot plant scale. 

2.      Line 90-Line 272. The content of Introduction is a bit long, can it be simplified?                  

Yes, the content of the introduction has been shortened (details in the text) without any significant reductions in the content provided.                         

      3.     and  4. Table 4, Table 5, Table 6. The content of these tables is substantial, but confusing, it is best to redesign.     The data comparison of BCPI, UA and TIA involved in this paper should    have significant analysis.

Tables 4, 5 and 6 have been reformatted with relevant comments referring to the statistical evaluation of the published research results. Details in the text. 

English was moderated.

Round 2

Reviewer 2 Report

I think the author has made serious changes. The manuscript has reached a level that can be published.